# The Barriers to and Facilitators of Physical Activity and Sport for Oceania with Non-European, Non-Asian (ONENA) Ancestry Children and Adolescents: A Mixed Studies Systematic Review

**DOI:** 10.3390/ijerph191811554

**Published:** 2022-09-14

**Authors:** Louisa R. Peralta, Renata L. Cinelli, Wayne Cotton, Sarah Morris, Olivier Galy, Corinne Caillaud

**Affiliations:** 1School of Education and Social Work, Faculty of Arts and Social Sciences, The University of Sydney, Sydney, NSW 2006, Australia; 2Charles Perkins Centre, The University of Sydney, Sydney, NSW 2006, Australia; 3Faculty of Education and Arts, Australian Catholic University, Strathfield, NSW 2135, Australia; 4Sydney Conservatorium of Music, Faculty of Arts and Social Sciences, The University of Sydney, Sydney, NSW 2006, Australia; 5Interdisciplinary Laboratory for Research in Education, University of New Caledonia, Nouméa 98800, New Caledonia; 6Faculty of Medicine & Health, The University of Sydney, Sydney, NSW 2006, Australia

**Keywords:** ONENA, PICTs, sport, physical activity, barrier, facilitator

## Abstract

Background: Participation in sport and physical activity (PA) leads to better overall health, increased life expectancy, and decreased mortality rates across the lifespan; however, there may be a range of individual, family, and community factors that influence PA participation among ONENA children and adolescents residing in the 22 Pacific Island Countries and Territories (PICT) and Australia. This review aimed to synthesise existing quantitative and qualitative literature regarding barriers to and facilitators of PA and sport among ONENA youth. Methods: The literature was systematically searched to include studies reporting barriers to and facilitators of PA and sports participation among ONENA children and adolescents aged 0–18 years residing in the 22 PICT and Australia. Using a pre-established taxonomy based on the social-ecological model, a deductive analysis was performed. Quality appraisal was performed using the mixed methods appraisal tool. Results: Of 1388 articles, 14 studies were included, with 128 ONENA children and adolescent participants across the four qualitative studies; 156,581 ONENA children and adolescents across the seven quantitative studies; 801 parents, children, and adolescents in one quantitative study; and 642 parents in two quantitative studies. Of the 14 included studies, none were based in Australia and only 10 of the 22 PICT were reported as the participants’ residence: Palau, New Zealand, Tonga, Cook Islands, Kiribati, Samoa, Solomon Islands, Tuvalu, Vanuatu, and Fiji. Four studies reported barriers, and another four studies reported facilitators of PA and sport, with the remaining studies reporting both barriers and facilitators. Overall, there were more barriers reported (30 in total) than facilitators (27 in total). Conclusions: Research in this area is lacking, with ONENA youth living in Australia and 12 PICT not represented. Overall, there were a larger number of facilitators experienced at individual and interpersonal levels, while barriers were highest at the community level, with the policy level having facilitators and barriers equally represented. Programs that offer PA and sport participation options with embedded SDT-informed strategies for all family members; that are accessible through existing transport and related social, cultural, and physical infrastructure; and that are committed to communities through formal co-design partnerships are needed, to enhance the PA and sport participation of ONENA youth residing in PICT.

## 1. Introduction

Engagement in regular physical activity (PA) is associated with better overall health, increased life expectancy, and decreased mortality rates lifelong [1,2,3]. Regular PA reduces the risk for cardiovascular disease [4], Type II diabetes [5,6], certain type of cancers [7], and obesity [4,5,6,7,8]. Moreover, engagement in regular physical activity is associated with better mental health outcomes [9].

Children and adolescents should do at least an average of 60 min of moderate-to-vigorous intensity PA per day, in addition to muscle strengthening PA at least 3 days a week [10]. However, research on children and adolescents’ PA shows that the overall PA levels tend to be lower than PA recommendations, decline with age, and are substantially lower for girls and those from low socioeconomic status (SES) groups [4,11,12]. As such, it is essential to continue the research on the factors that facilitate or inhibit PA in adolescent populations. A recent updated systematic review of the qualitative literature on adolescents’ perspectives of the barriers to and facilitators of PA (a total of ~1250 adolescents from 13 countries), reported five influencing factors: (1) individual factors (e.g., psychological—motivation, self—efficacy; cognitive—knowledge, understanding; physical—motor skills); (2) social and relational factors (e.g., family, friends, significant others); (3) PA nature factors (e.g., fun, school-based PA and physical education); (4) life factors (e.g., time and competing activities; life-course); and (5) sociocultural and environmental factors (e.g., availability/access to PA facilities, programs; urban/rural zones) [13].

Several other notable reviews have also contributed to the study of PA in adolescent populations, further clarifying the barriers to and facilitators of PA in specific youth demographics [13,14,15,16,17]. In brief, some of the prevalent PA barriers identified across these four reviews were related to: (1) lack of fun, motivation, and perception of competence; (2) body image and gender bias in sport and PA; (3) lack of support from family, friends, and significant others; (4) negative experiences in PA and physical education contexts; (5) competition and highly structured PA opportunities; and (6) limited environmental opportunities. Conversely, the primary facilitators of PA in young people were reported to be (1) positive attitudes towards PA; (2) fun, motivation, and perception of competence; (3) positive perception of body image and challenging stereotypes; (4) friends, family, and significant other support for PA; (5) positive experiences in physical education and PA; (6) a safe environment; and (7) access to PA programs and recreational infrastructures. However, these reviews have focused on only specific youth demographics, such as those residing in the UK [14,15], young females [16], children with disabilities [17], or from urban contexts and high-income economy countries [13]. The literature from low- or middle-income economy countries is sparse, especially in the Pacific region.

A substantive amount of research on the facilitating or hindering factors of PA participation has come from correlates and determinants in quantitative approaches [18,19,20,21]. By combining the findings of qualitative studies in reviews, quantitative research can canvas a comprehensive overview of the essential elements to fully capture the extent and depth of the complexity of PA behaviour, and its interdependence within and across different levels and contexts. Together, qualitative and quantitative data will be able to inform and update further large-scale quantitative studies and scalable interventions.

Populations living in the Pacific are generally described using the generic and vague terminology “Pacific Islanders” and are delineated into three main geographic areas: Micronesia, Polynesia, and Melanesia. Due to the geographical dispersion in the south Pacific regions and recent influences from either Anglo-Saxon or French colonisations (post World War 2), there is incomplete information on the lifestyles, health and wellness of people living in the south Pacific region. Misrepresentations of Pacific Islanders in cohort studies and epidemiological surveys have led to persistent underestimations of their population size. However, if we consider all people living in Oceania with Non-European, Non-Asian (ONENA) ancestry, then there is a population of 11.6 million people across 22 Pacific Island Countries and Territories (PICT) [22]. While we recognise the diversity of populations and cultures living in the Pacific and the need to consider communities in their context, with their own knowledges and practices, it is also important to recognise that, as a whole, the ONENA people represent a very large group (see Figure 1).

Over the past two decades, the region has experienced a major shift in disease burden: non-communicable diseases (NCDs) have overtaken communicable diseases, with the prevalence of NCD risk factors in the Pacific region ranking among the highest in the world [24,25,26]. The leading NCD risk factors in the Pacific Island region have been identified as unhealthy diets, physical inactivity, tobacco use, and alcohol misuse [27,28,29]. While NCDs account for 60% of global deaths [30], NCDs account for ~75% of all recorded deaths in the Pacific region [27]. A significant proportion of NCD morbidity, disability, and premature deaths within the region could be prevented through population-based PA interventions, even though current prevalence rates of PA among the ONENA population are not reported [31]. In 2012, a stocktake of PA programs in the PICT’s, found 84 PA initiatives had been conducted in 20 PICT’s, with 17 of these initiatives implemented in a school setting [31]. However, an evaluation of program efficacy found that these school-based PA programs were limited by low participation rates and reach. Only two studies conducted in New Caledonia showed the effectiveness of a digital PA intervention among ONENA adolescents [32], as well as the influence of socioeconomic status, place of living (e.g., tribal, rural, urban), ethnicity (European, ONENA, Asian), and safety in urban areas on PA for adolescents [32,33,34,35]. Therefore, little is known about the barriers to and facilitators of PA faced by ONENA adolescents residing in PICT and Australia.

There may be a range of individual, family, and community factors that can influence participation in PA for ONENA children and adolescents [29,32,36]. Various models such as the social-ecological model have been used to explore barriers to and facilitators of PA [4], as the model explores the interactive effects of individual, interpersonal, institutional, and community features, as well as public policy, on behaviour. In a narrative review conducted by May [37], a social-ecological model was used to further evaluate how individual engagement in PA was influenced by cultural and environmental factors, and vice versa. For example, macro-social factors including colonialism, discrimination, and dispossession were identified as negatively impacting on Aboriginal and Torres Strait Islander health. Conversely, at the individual level, some urban Aboriginal and Torres Strait Islander young people reported that they used PA to manage stress. Differing cultural views on PA and exercise were identified: From a Western perspective, the concept of exercise was related to an individual taking care of their health, whereas engagement in PA for Aboriginal and Torres Strait Islander People was more related to social roles and communal activities [38]. Economic and social disadvantage were identified as limiting opportunities for young Aboriginal and Torres Strait Islander People to participate in sport and PA. While barriers and facilitators were not systematically explored in May’s [37] review, it does highlight the complex interaction of social-ecological levels that affect Aboriginal and Torres Strait Islander participation in PA [38] and provides insights that might be relevant for other first nation populations, such as ONENA people residing in the PICT.

The present systematic review aims to address the aforementioned gaps by (1) synthesising the findings of existing research that focused on the barriers to, and facilitators of, participating in PA and sport for ONENA children and adolescents (0–18 years of age) living in the PICT and Australia; and (2) evaluating the quality of the studies conducted. To the authors knowledge, this is the first published review to synthesise the barriers to and facilitators of sport/PA participation among ONENA children and adolescents residing in PICT, and of Australia (but not including Aboriginal and Torres Strait Islander people, due to recent reviews with this sub-population group [37,38]). The findings have the potential to inform practice through supporting the development of future experimental PA and sport studies and program evaluations designed with and for ONENA children and adolescents residing in PICT and Australia.

## 2. Methodology

### 2.1. Protocol and Registration

This systematic review was pre-registered with the International Prospective Register of Systematic Reviews (PROSPERO) (registration number: CRD42021290746) and followed the Preferred Reporting Items for Systematic Reviews and Meta-Analysis (PRISMA) guidelines [39].

### 2.2. Eligibility Criteria

To be included in the review, original research articles needed to be published in English in a peer-reviewed journal, with a protocol that followed an observational (e.g., cross-sectional, cohort, case control, qualitative) or experimental (e.g., acute or experimental randomized or quasi trial) study design. Our original search included articles published between 1 January 1946 (after World War 2) and 31 August 2021. Reviews, protocol papers and conference abstracts were excluded.

Additional eligibility criteria were established using Population, Intervention or Exposures, Comparator, and Outcome (PICO) study criteria [40]. Articles were included if they reported on populations in early childhood (birth through 6 years at baseline) and up to the end stage of adolescence (18 years of age). Articles were also included if adults (parents/family/community members and service providers) reported on barriers to and facilitators of PA on behalf of children and adolescent individuals (0–18 years of age). Exposures for observational studies included participation in physical activity (i.e., any bodily movement that resulted in energy expenditure), sport, leisure activities, or fitness (e.g., cardiorespiratory or muscular fitness). Exposures for experimental studies included acute bouts (i.e., a single session) of physical activity or a physical activity or fitness intervention. Comparators were defined as the non-exposed group (e.g., less physically active compared with more physically active) or the participant sample (e.g., qualitative study) in observational studies; and the standard care, alternative condition, or intervention group(s) in experimental studies. Outcome variables were objective or subjective measures of correlates, facilitators, barriers, and socioecological factors.

### 2.3. Information Sources and Search Strategy

A systematic literature search was conducted using Boolean strategies, with a predefined list of keywords (i.e., various terms for physical activity, cognition, school readiness skills, and youth) in PubMed, PsycINFO, Sports Discus, ERIC, MedlinePlus, and Scopus. A manual search was also conducted using citation chaining of the included studies. Our list of search terms was developed from a compilation of key terms used in PA and youth review papers [13,16,41] and through refinement during preliminary searches (e.g., search terms were assessed for comparability across databases). Specific search terms used were

Populations: “pacific island*”, “oceanic ancestry group”, “small island developing state*”, polynesia*, micronesia*, melanesia*, Pasifika* PLUS the name of the 21 PICs AND keywords kid* OR child* OR adolescent* OR student* OR “young people” OR “youth” OR teen* OR “high school” OR “middle school” OR minor* OR juvenile* AND Contexts: Sport* OR Recreation OR “Physical Activ*” OR Exercis* OR Fitness* OR “sport program*” OR “physical education” OR active OR movement OR leisure OR inactive AND Outcomes: correlate* OR determinant* OR facilitator* OR barrier* OR “factor influen*” OR “socio-ecological factors” OR “psychosocial factor*” OR “environmental factor*” AND Study methods: “evidence-based” OR effective* OR treatment* OR intervention* OR outcome* OR “experimental stud*” OR “quasi-experiment*” OR “case stud*” OR “case-control stud*” OR “cross-sectional” OR “cohort stud*” OR observational OR “promising practice*” OR “randomized control trial*” OR interview* OR “focus group*” OR narrative* OR qualitative OR survey OR “pre-experiment*” OR evaluation OR perspective* OR voice* OR experience* OR “grounded theory”.

Results from each database were imported into the reference manager software, EndNote X9 (EndNote. Endnote X9 Software; Clarivate Analytics: Philadelphia, PA, USA, 2013). These references were then imported into the online review platform, Covidence (Veritas Health Innovation, Melbourne). After duplicates were automatically removed, five authors independently screened the titles and abstracts of all included studies. A full text document was obtained for each article that met the initial screening criteria and then independently reviewed by two authors. Conflicts resulting during these stages were resolved either via consensus or by an independent reviewer. Literature was excluded during full text screening if full texts were not available.

### 2.4. Data Extraction

Two independent reviewers used a standardised form to extract methodological, demographic, and results data. Data were extracted on child/youth characteristics (number of participants, participant age range, gender), the location and funding of studies, study aims and outcomes, methods, and reported barriers and facilitators. Due to the inclusion of studies that used qualitative and quantitative designs, a meta-analysis of the data was deemed inappropriate. A descriptive narrative synthesis was chosen as the most relevant and suitable method of data synthesis for this review.

Data were extracted using the social-ecological model proposed by May [37], which categorised barriers to and facilitators of PA into four levels (using the socio-ecological framework): (1) Individual: characteristics including, but not limited to, personal attitudes, motivations, and self-efficacy; (2) Interpersonal: family, friends, peers, and other social support systems, as well as connection with social and cultural practices; (3) Community: the provision of sport and PA programs, facilities, resources, and transport. Climate, safety, and environment factors were also included in this category; and (4) Policy: government policy and programs, and community led programs.

The following characteristics were extracted by two independent authors: (1) first author’s name, publication year, country, and funding; (2) study aim; (3) study design and theoretical framework; (4) sample characteristics; (5) data collection and analysis procedures; and (6) study results. For the results section, a thematic analysis was conducted to determine the main themes that had been reported as barriers to and facilitators of PA [42]. This analysis involved six phases: (1) familiarisation with the data; (2) generation of initial codes; (3) identification of themes; (4) refinement of themes; (5) definition and naming of themes; and (6) production of the report. Inductive coding was undertaken by the first author and RC. A coding framework was developed by synthesising the code lists of the two independent authors, with any discrepancies between codes being resolved via author consensus. Any units of texts that were not coded during this process were discussed by the two authors, with new codes being created if necessary, thus refining and expanding the coding framework. Deductively, codes were placed under the relevant level within the social-ecological framework.

### 2.5. Quality Appraisal

Two independent reviewers critically assessed the methodological quality and risk of bias of the included studies using the mixed methods appraisal tool (MMAT) [43]. The MMAT was chosen for this review as it is appropriate for the appraisal of the included qualitative, quantitative, and mixed methods studies. This appraisal included two screening questions for all study types and five questions for each individual study design. These questions assessed the quality of the included studies through the evaluation of the study aims and designs, recruitment strategies, data collection methods, data analysis methods, presentation and discussion of findings, and final conclusions. Studies were scored using percentages (0–100%). Any discrepancies were discussed until a consensus was reached.

## 3. Results

### 3.1. Study Selection

Database search results included 2152 studies. These studies were imported into Covidence (Clarivate Analytics), where 764 duplicates were then automatically removed. The remaining 1388 studies were then screened according to the title and abstract for relevance, resulting in another 1278 studies being eliminated. The full texts of the remaining 110 studies were then screened, leading to another 96 studies being excluded. The main reason for exclusion was that most studies (n = 68) did not report on ONENA children and adolescents. Other reasons for exclusion are identified in Figure 2. After screening, 14 studies were selected for final data extraction and risk of bias assessment.

### 3.2. Study Characteristics

Table 1 shows the characteristics of all 14 included articles, which represent studies published between 2006 and 2020 that reported on the barriers of and facilitators to participating in PA and sport for ONENA children and adolescents (0–18 years of age) living in the PICT and Australia. Most studies measured and reported barriers and facilitators using qualitative methods (n = 9) [44,45,46,47,48,49,50,51,52] and thematic analysis (n = 4) [53,54,55,56]. The remaining study did not disclose its analysis method [57].

### 3.3. Participants

There were 134 participants in total, which included 128 ONENA children and adolescent participants, across the four qualitative studies (the other 6 participants were research intervention managers and assistants) [56]; 156,581 ONENA children and adolescents across the seven quantitative studies [44,45,46,47,49,51,52]; 801 parents, children, and adolescents in one quantitative study (study did not report separate sample participant numbers for parents and children) [50]; and 642 parents in two quantitative studies [47,48]. Of the 12 studies with children and adolescent participant samples [44,45,46,47,49,50,51,52,53,54,55,57], only one included participants less than 11 years of age (8–12 year old participants; [53]). Only 10 of the 22 PICT were reported as the participant locations of residence, including: Palau [57], New Zealand [45,46,47,48,53,54], Tonga [44,49,51], Cook Islands [49], Kiribati [49,51], Samoa [49,50,51,52], Solomon Islands [49,51], Tuvalu [49,51], Vanuatu [49,51,55], and Fiji [51,56].

Six studies were conducted in New Zealand, which is considered a high-income country (Countries and Economies. Available online: https://data.worldbank.org/country (accessed on 31 January 2022)), with one other study being conducted in Palau, another recognised high-income country (Countries and Economies. Available online: https://data.worldbank.org/country [accessed on 31 January 2022]). The remaining studies were conducted in lower income countries, including two studies with upper-middle income economies (Tonga and Fiji), three with lower-middle income economies (Samoa, Solomon Islands, and Vanuatu), and two studies with lower income economies (Tuvalu and Kiribati) (Countries and Economies. Available online: https://data.worldbank.org/country [accessed on 31 January 2022]).

### 3.4. Quality Appraisal

Table 2 shows the quality appraisal ratings of the included studies. All the studies scored over a 60% quality rating, with ten of the fourteen studies scoring 100% [44,45,46,47,48,49,51,52,54,55]. The remaining four studies scored 60% (n = 2) [50,56], and 80% (n = 2) [53,57]. Overall, the included articles were considered moderate–high quality.

### 3.5. Barriers and Facilitators

Table 3 displays the barriers and facilitators using the social-ecological model (See Methods). Four studies reported barriers to [44,45,49,56], and facilitators of [47,48,52,55] PA and sport. There were seven individual, six interpersonal, 15 community, and two policy level barriers; and nine individual, ten interpersonal, six community and two policy level facilitators. Overall, there were more barriers reported (30 in total) than facilitators (27 in total). When focusing on studies conducted in high income countries only [45,46,47,48,53,54,57], the barriers outnumbered the facilitators (19 to 16). The studies conducted in middle to high income countries [44,56] highlighted two facilitators and no barriers, with low to middle and low-income countries reporting a greater number of barriers compared with facilitators (11 to 10).

At the individual level, the facilitators outnumbered the barriers (9 to 7) and were mainly derived from two studies [53,54]. Fun and engaging physical activities were the most popular individual facilitators, as reported by children, adolescents, and their parents [53,54]. These activities were preferred over organised sport or structured activities [53]. The other six facilitators included preferential activities (e.g., preferred to be active rather than engaged in household chores), positive self-esteem, feelings of achievement, physical health benefits (i.e., gaining strength), improved physical appearance, and mood and confidence. These facilitators were all reported by the same two studies [53,54]. Lack of time and energy, low interest, discomfort (e.g., sweatiness), and transportation difficulties were the most common barriers. These barriers were reported in six of the 14 studies [45,46,48,50,52,53].

At the interpersonal level, the facilitators outnumbered the barriers (11 to 6). Facilitators were reported in eight studies [45,48,49,50,51,54,56,57], and barriers were reported in five studies [47,48,50,51,53]. The most important facilitators for ONENA children and adolescents were parental support and involvement in physical activity decisions [48,49,50,51], and the promotion of physical activity from friends and other role models [50,54,57]. Other facilitators included walking to school [45] and the connection of physical activities with cultural practices (e.g., traditional dance or involvement of the church) [50,56]. The barriers were, in some cases, similar to the facilitators. Two studies reported that having inactive friends, no friends, or being bullied by peers were barriers to being engaged in PA and sport [48,51]. In another study, cultural practices that promoted activity were also seen as a barrier [50]. Other barriers included financial struggles, difficulties in getting to PA venues [53], and heavy schoolbags that restricted active transportation [47].

At the community level, there were 15 reported barriers and six facilitators. Barriers at the community level included limited access to indoor sporting facilities [57], distance between sporting facilities and living/household locations [54], inadequate footpaths and cycle lanes [48,50], and limited organised or structured activities for all age-groups within the population [54,57]. Other barriers included safety concerns, such as general safety within communities and on the road [45,48,50,53,54], and a fear of dogs and dog bites [50,52]. Physical environment barriers included the hot climate [45,57]. The facilitators for PA and sport at the community level included PA support from medical practitioners [50], integration of sociocultural factors into PA and sporting programs (e.g., traditional dancing) [44], and supportive community and cultural leaders that provide youths with physically active role models [50,53,55].

At the policy level, there were two barriers to and two facilitators of PA and sport. The most common facilitators included the continuity of PA and sport program provision [54], and physical education classes being scheduled five days/week during school hours [51]. In contrast, food insecurity [51] and village curfews [50] were barriers for regular engagement in PA and sport.

## 4. Discussion

This review synthesised the current literature focusing on the barriers to and facilitators of PA and sports participation for ONENA children and adolescents living in PICT. There were no studies that focused on ONENA children and adolescents (excluding Aboriginal and Torres Strait Islander children and adolescents) residing in Australia and barriers to and facilitators of engagement in PA and sport. Half of the studies were conducted in high income countries such as New Zealand (n = 6) and Palau (n = 1), with the remaining half from countries with lower-medium income economies, as defined by the World Bank (Countries and Economies. Available online: https://data.worldbank.org/country (accessed on 31 January, 2022)). This study focused on the identification of barriers to and facilitators of PA for ONENA youth living in PICT at each level of the social-ecological model. From this process, there were a lower number of total facilitators, compared with barriers (27 versus 30) reported across the included studies. However, there were a larger number of facilitators experienced at the individual and interpersonal levels, with barriers highest at the community level. At the policy level, facilitators and barriers were equally represented.

Fun and engaging physical activities, which are non-organised/unstructured, were the most popular individual level facilitators that would encourage ONENA children and adolescents to engage in regular physical activity, as reported by children and adolescents themselves, as well as their parents. This finding is similar to a small group of studies within Martins and colleagues’ [13] updated systematic review of qualitative studies reporting barriers to and facilitators of PA for adolescents. Three of the 30 studies included in the Martins et al. [13] review noted that adolescents found that informal/unstructured and inclusive PA experiences facilitated engagement in PA [58,59,60], due to the activities being new, diversified, adventurous, and fun. Other individual level facilitators were experienced when the children and adolescents were engaged in PA and sport, and therefore several PA benefits were noted related to physical and mental health, including positive self-esteem, feelings of achievement, improved body image, fitness, strength, and mood. These were identified by children and adolescents in eight studies in Martin’s [13] systematic review [61,62,63,64,65,66,67,68], thus highlighting that children and adolescents’ motivational profiles are an important consideration for facilitating PA and sport and that educating young people for a lifetime of PA needs to consider motivation and meaning [69]. The application of motivational theories, such as self-determination theory (SDT), in PA or sport program development is encouraged, where there is growing evidence that, when implemented appropriately, SDT-informed pedagogical approaches can empower young people’s basic psychological needs (i.e., autonomy, competence, and relatedness) and the promotion of democratic and inclusive PA and sport experienced [70,71,72]. The most reported barriers were related to low interest and discomfort (due to sweatiness and clothing), as well as lack of motivation and time. Even though the motivational barriers were addressed through SDT-informed practices, the additional barriers to participation in PA identified (e.g., discomfort) show how the complex interactions of individual and cultural factors can be strong determinants of the personal behaviour of individuals. May’s [37] mixed studies systematic review, focusing on determining the barriers to and facilitators of physical activity and sports participation in Aboriginal and Torres Strait Islander children, highlighted the need to consider the specific cultural milieu and contexts of individuals. This also applies to ONENA children and adolescents residing in PICT. Each PICT presents its own cultural context, and to ensure that PA and sport programs are suitable, designers and practitioners need to find out from ONENA young people in each of these contexts where and what kinds of PA they are most comfortable participating in, and the appropriate timing during the day and session length, as well as suitable clothing. Furthermore, not having enough time to regularly do PA was related to the time required for family and cultural duties and other conflicting obligations. This is supported by several studies in Martins’ [13] review, where family duties were increased for those young people from ethnic minorities or those living in lower to middle income economy countries [63,64,65,68,73].

Family and friends who were active was one of the most reported facilitators at the interpersonal level; and if they were inactive this was a barrier. This highlights the importance of participation in PA and sport for ONENA children and adolescents, and their respective family members; in that they act as role models. This was also supported by May’s [37] review, which highlighted the need for adult level barriers and facilitators to be considered in future research, due to their influence on young people. For example, providing family PA and sporting opportunities, and adult sport and PA options, where parents and other family members can interact and participate together, provides support and encouragement. At the same time, transportation barriers will be overcome. In addition, ONENA young people in PICT are more likely to engage in PA and sport if the PA and sport experience have strong connections with cultural practices, such as traditional dance or involvement of the church. Addressing sociocultural factors in PA and sport programs (such as traditional, cultural dance) in PICT and having community and cultural leaders as role models and facilitators is an important recommendation for overcoming interpersonal and community level barriers and enhancing facilitators.

At the community level, barriers far outweighed facilitators. The barriers to participation in PA and sport included lack of access to sporting facilities, large distances between sporting facilities and residential areas, inadequate footpaths and cycle lanes, reduced number of organised PA opportunities and sport programs, and transport to and from venues. Living in a “hot and sweaty” climate was an additional barrier for many ONENA young people living in PICTs. Again, this finding is supported by Martins et al.’s [13] and May et al.’s [37] review, as well as a recent study aiming to determine the effects of heat on children and early adolescents’ physiological responses [74], in which there are several physical and physiological barriers experienced by young people who live in hot or tropical climates. Another barrier experienced by ONENA youth was safety, which led to the facilitators that highlighted the importance of parent and community member engagement with youth in PA and sport. This finding is supported by another study conducted with adolescents living in rural and urban New Caledonia (a PICT), which indicated that safety was an important driver for engagement in PA, particularly for younger children and adolescent females [35].

Adolescents living in Samoa voiced, not only general safety and road safety as barriers, similar with other international research focusing on adolescents and PA [75,76], but fear of dogs and dog bites [50,51,52]. Few studies have focused on dogs and dog bites in decreasing or limiting adolescents’ PA levels outside of Samoa, and therefore future PA promoting programs in Samoa should consider this barrier and design alternatives to engaging in PA in external locations for adolescents (e.g., use internal community facilities).

It is not surprising that the facilitators for PA and sport at the community level included medical and doctor support, addressing sociocultural factors in PA and sport programs, and having community and cultural leaders as role models and facilitators. Using these facilitators, future PA and sport programs for ONENA youth in PICT need to consider co-design processes, implement PA and sport programs that are readily accessible for all community members (not just youth) to improve safety and enhance role modelling, accessed by medical support to encourage community members to participate safely, and are connected to a central location that celebrates culture (e.g., church).

The continuity of programs and village curfews were a barrier, and, as such, program provision with curfew-free times was a facilitator, even if these programs were designed and implemented by others. This contrasts with May et al.’s [37] findings that “welfarism”, where communities were disempowered by external organizations’ coordinating programs, was a barrier to PA and sport engagement in Australian Aboriginal communities. Another facilitator for PA at the policy level was if PE was offered five days/week during school hours, with this supported by several international studies [13,77,78]. Other barriers to engagement in PA and sport included food insecurity. As shown in several recent studies, food insecurity reduces PA engagement for youth and families [79,80,81]. In Fram and colleagues’ [79] study, children with the highest level of food insecurity participated in 17 min/d less PA compared with those children who were food secure. This was supported in To and colleagues’ [81] study, where they reported that food insecure children did less moderate to vigorous PA, and food insecure adults were less likely to adhere to PA guidelines than those without food insecurity. In Gunter et al.’s [80] study, after adjusting for education, race, ethnicity, and eligibility for federal meal programs, readiness to encourage and provide opportunities for PA was significantly lower among food insecure families who resided in rural areas. Therefore, future PA and sport program designers and practitioners should consider food security, especially when co-designing with communities who reside in rural, remote, and/or low to middle income countries, and when determining program designs and assessment of child health and well-being.

Although every effort was made to ensure the rigour of this mixed-methods systematic review, by registering the review with PROSPERO, using the PRISMA guidelines [39] and an established quality appraisal tool [43], as well as an ecological theoretical framework to capture facilitators and barriers (as used in previous research [13,37]), future research focusing on physical activity promotion with youth may need to consider adopting the social determinants framework proposed by the Centers for Disease Control and Prevention (CDC) Healthy People 2020 [82]. The social determinants framework categorizes facilitators and barriers via the social determinants of health (i.e., economic stability, education, food, neighbourhood and physical environment, community and social context, and healthcare system) [82]. As the social determinants are specific and contextually relevant across and within countries, each determinant can be explored in greater depth individually and then collectively to capture the interactions and complexity of youth PA promotion. 

## 5. Conclusions

Programs which offer PA and sport participation options with embedded SDT-informed strategies for all family members; that are accessible through existing transport and related social, cultural, and physical infrastructure; and are committed to communities through formal partnerships are needed. There are unique barriers in PICT including climate, safety, and food security factors, and a lack of (ongoing) sporting and PA programs, which will need targeted approaches to overcome.

## Figures and Tables

**Figure 1 ijerph-19-11554-f001:**
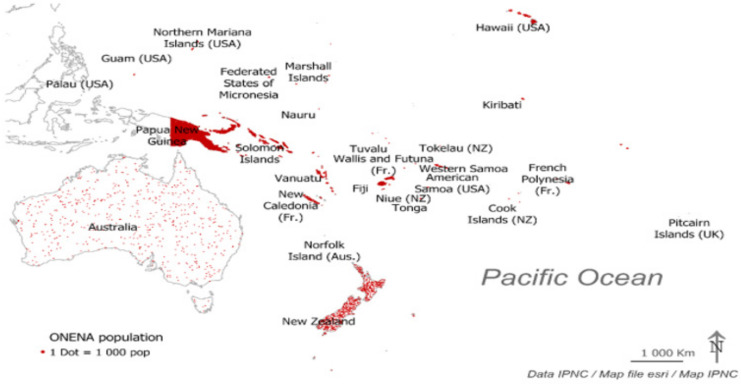
Oceania and relative estimates of population subgroups. Adapted from [23].

**Figure 2 ijerph-19-11554-f002:**
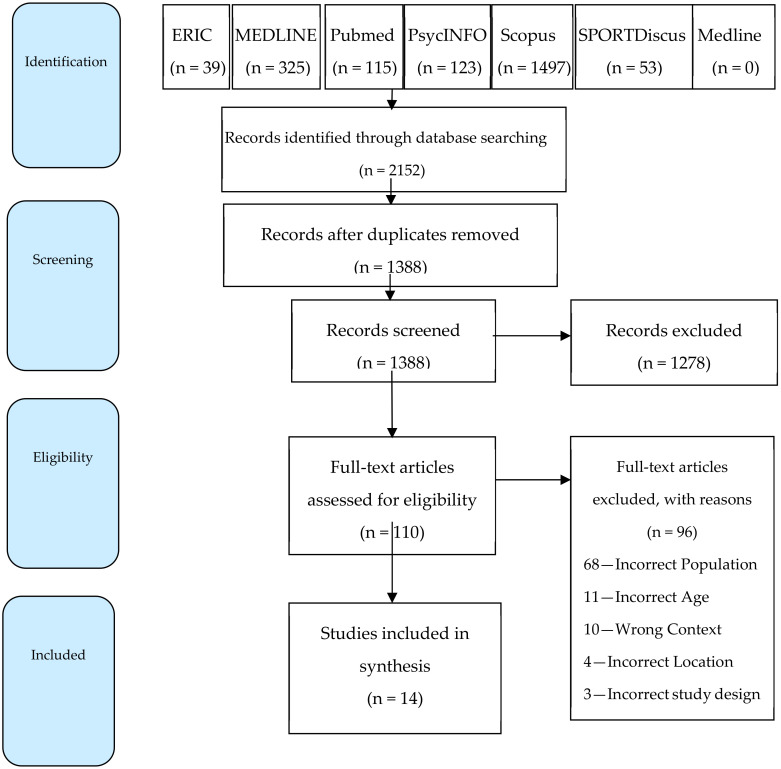
PRISMA flow diagram for study selection.

**Table 1 ijerph-19-11554-t001:** Summary of the included qualitative and quantitative studies reporting barriers and facilitators associated with sport/PA participation.

Author (Year), Country, Funding Agency	Aim	Design/ Theoretical Framework	ONENA Sample (Sex, Age, Grades)	Data Collection and Analysis	Key Findings as Reported in Papers
Collier, A.F. et al. [57], Palau, NR.	To establish an advisory council of local stakeholders. Conduct an extensive needs assessment with youth, parents, professionals, and the lay public in Palau to inform a wellness intervention.	Qualitative study with cross-sectional data collected (CBPR model).	The mean age of participants was 30.6 years; 38.5% of the sample was under 18 years of age. The majority of the 43 youth and adults in the sample were Palauan (81%).	-Study 1 examined the reasons for overeating in Palau; the best methods of service delivery for the program; and the key features for the wellness program. The sample included youth and adults. -Study 2 examined rates of ow/ob, eating disorder symptomatology, exercise, and the patterns and types of food eaten by Palauan youth. -Data analysis was not reported in the paper.	Barriers to PA: -The average air temperature in Palau is 82 °F (28 °C) and the average relative humidity 82%. Due to the excessive heat, it was indeed challenging for participants to find exercise activities that they wanted to do. -Excessive heat and humidity make it difficult to walk outside during daylight and there are few options for indoor exercise. Facilitators for PA: -More work is needed to brainstorm with community members of different ages the types of physical activity that they can do, as well as what time of day works best for them. -Working with participants to design physical activities that can be completed inside at home or work would also be beneficial. -More emphasis on finding individual partners for physical activity would be helpful for compliance.
Curtis, A.D. et al. [53], New Zealand, NR.	To determine which factors influence children from areas of socioeconomic deprivation to engage in after school activities. Findings intended to provide a cross-sectional study basis for developing future after school physical activity programs in these areas.	Qualitative study with cross-sectional anthropometric data collected (e.g., BMI)/NR.	Nine children (age range ~8–12 years old) and 21 parents (age range ~31–43 years old) participated in the study; 38% of the sample identified as Pacific Islander (28%) or Maori (10%).	-Focus groups with children, utilising photo-voice prompts for discussion. -Focus groups and semi-structured interviews were conducted with parents. -Content analysis of data was undertaken.	Barriers to PA: -Parents perceived that time, money, and transport were all barriers to children participating in physical activities after school. Facilitators for PA: -Both children and parents described physical activity and play as different constructs; physical activity was considered as an organised activity and play was identified as fun. -Parents explained that children’s enjoyment of a particular activity, as well as positive self-esteem, influenced children’s participation in physical activity. -Community support and communication were also identified as important in creating safer communities and places to play for children.
Fotu, K.F. et al. [44], Tonga, Welcome Trust (UK), the National Health and Medical Research Council (Australia) and the Health Research Council (New Zealand) through their innovative International Collaborative Research Grant Scheme.	This paper presents the results of the Ma’alahi Youth Project (MYP), the first community-based intervention to target adolescent obesity in the Kingdom of Tonga.	The Ma’alahi Youth Project, the Tongan arm of the Pacific Obesity Prevention in Communities project, was a 3-year, quasi-experimental study of community-based interventions among adolescents in three districts on Tonga’s main island (Tongatapu) compared to the island of Vava’u/CBPR	The intervention group comprised 815 secondary school students aged 11–19 years from the districts of Houma, Nukunuku, and Kolonga on the main island of Tongatapu. The comparison group comprised 897 secondary school students aged 11–19 years from the island of Vava’u.	Anthropometric data, including height, weight, waist circumference, and body fat percentage. Behavioural and quality of life survey data were collected. Health-related quality of life was measured using two instruments: the Assessment of Quality-of-Life instrument (AQoL-6D) and the Paediatric Quality of Life Inventory 4.0 (PedsQL; generic module for 13- to 18-year-olds).	Barriers to PA: Negative outcomes relating to physical activity with a smaller proportion of intervention participants walking/riding to or from school (*p* = 0.001), being active at lunchtime (*p* = 0.001), and engaged in after-school activity (*p* = 0.008) than comparison group participants at follow-up. Facilitators for PA: -Greater integration of strategies to address socio-cultural factors under-pinning food and physical activity patterns, as well as body size perception, into the intervention may have strengthened the “dose” of the overall intervention and led to more beneficial outcomes.
Hohepa, M. et al. [54], New Zealand, Health Research Council of New Zealand and Auckland University of Technology.	To explore the views of school students on various physical activity contexts and their ideas for potential physical activity promoting strategies.	Qualitative study/socio-ecological model.	In total, 44 participants took part in focus group discussions: 24 females and 20 males (age = 13–15 years). Maori participants comprised just over 50% of the female (n = 13) and male (n = 11) sub-samples.	Nine focus group sessions. The focus groups were separated according to ethnicity (Maori and New Zealand European) and gender, and included a maximum of six participants (range 3–6) in each group. Data analysis included thematic induction using the long table approach.	Barriers to PA: -Six major themes relating to supportive sedentary environments (e.g., passive transport, accessibility and availability, electronic devices), peer influences, structure of PA opportunities, physical constraint (e.g., distance, safety), motivation level, and lack of time (e.g., home/school/work duties). Facilitators for PA: -Five major themes relating to fun, achievement, physical (e.g., health benefits, physical appearance), psychological (e.g., mood and confidence), and preferential activity (e.g., get away from domestic duties/expectations) factors. Potential physical activity promoting strategies: -Identified by high school students: Availability and accessibility, peer and familial support, and self-responsibility.
Mandic, S., Hopkins, D., et al. [45], New Zealand, National Heart Foundation of New Zealand, Lottery Health Research Grant, University of Otago Research Grant, Dunedin City Council.	The aim of the study was to compare correlates and perceptions of walking versus cycling to school in Dunedin adolescents living less than 4 km from school.	Cross-sectional study/socio-ecological model	Adolescents (n = 764; 44.6% males; 13–18 years; mean age 15.2 years ± 1.4 years) from 12 secondary schools. Maori participants = 9.3%.	Participants completed an online survey about perceptions of walking and cycling to school. Distance to school was calculated using Geographic Information Systems network analysis. Variables assessing perceptions of walking versus cycling using 4-point or 7-point Likert scales were analysed as continuous variables using paired *t*-tests. To calculate the proportion of adolescents agreeing with each statement, 4-point Likert scale items were also recoded into “disagree” and “agree” and 7-point Likert scale items were recoded as “disagree”, “neutral”, or “agree”. Data analysis was performed using SPSS Statistical Package (Version 22). To account for multiple tests, a *p*-value of 0.001 was chosen to indicate statistical significance.	-Overall, 50.8% of adolescents walked and 2.1% cycled to school, 44.1% liked cycling for recreation and 58.8% were capable/able/confident to cycle to school. Barriers to PA: -Compared to walking, adolescents reported that cycling to school was perceived as less safe by themselves (cycling vs. walking; 61.3% vs. 89.8%) and their parents (71.4% vs. 88.6%) and was less encouraged by their parents (23.0% vs. 67.0%), peers (18.8% vs. 48.4%), and schools (19.5% vs. 30.8%) (all *p* = 0.001). -The route to school had fewer cycle paths compared to footpaths (37.2% vs. 91.0%; *p* = 0.001). -Compared to walking, cycling to school provided less opportunity for socialising with friends (*p* < 0.001) and posed more personal barriers (e.g., too much to carry, after school schedule, need for planning and getting sweaty) (<0.001). Distance to school (*p* = 0.189) and wet and cold weather (*p* = 0.845) were barriers for both walking and cycling. Facilitators for PA: -Adolescents expressed more positive experiential (walking: 45.9%; cycling: 34.9%) and instrumental beliefs (walking: 74.2%; cycling: 59.2%) towards walking versus cycling to school (*p* = 0.001). Potential physical activity promoting strategies: -Cycle friendly uniforms (41.4%), safer bicycle storage at school (40.1%), slower traffic (36.4%), bus bicycle racks (26.2%), and bicycle ownership (32.7%) would encourage cycling to school.
Mandic, S., Sandretto, S., et al. [46], New Zealand, National Heart Foundation of New Zealand, Lottery Health Research Grant, University of Otago Research Grant, Dunedin City Council.	This study examined correlates of adolescents’ enrolment in the closest school in the absence of school zoning policies.	Cross-sectional study/socio-ecological model	Adolescents (n = 797; age: 15.2 ± 1.4 years; 51.4% boys) from six non-integrated (regular) public secondary schools without school zoning in Dunedin, New Zealand. Maori participants = 12.7%.	Participants completed an online survey about school choice. Distance to school was calculated using Geographic Information Systems network analysis. Data were analysed using *t*-tests, Chi-square tests and mixed effects binary logistic regressions.	Facilitators for PA: Overall, 51.3% of adolescents enrolled in the closest school (range across schools: 28.3% to 81.6%). Adolescents enrolled in the closest school had five-times higher rates of active transport (46.5% vs. 8.8%) and lower rates of motorised transport to school (40.3% vs. 68.8%) compared to their counterparts (all *p* < 0.05).
Mandic, S. et al. [47], New Zealand, Health Research Council of New Zealand Emerging Researcher First Grant, the National Heart Foundation of New Zealand, a Lottery Health Research Grant, a University of Otago Research Grant, the Dunedin City Council.	This study examined parents’ and adolescents’ perceptions of school bag weights and actual school bag weights for adolescents in New Zealand.	Cross-sectional study/socio-ecological model	Parents (n = 331; 76.7% women; 6.0% Maori) and adolescents (n = 682; age 15.1 SD 1.4 years; 57.3% boys; 10.9% Maori).	Parents and adolescents completed the BEATS Study Parental or Student Survey, respectively. Survey questions were related to demographics (age, gender, ethnicity), travel to school behaviours, and perceptions of walking and cycling to school. Home and school addresses were geocoded (converted into coordinates), then used to calculate distance to school using the shortest path on a connected street network using geographic information system (GIS) network analysis. Height (custom-built portable stadiometer), weight (A&D scale UC321, A&D Medical, San Jose, CA, USA), and waist circumference were collected from adolescents.	Barriers to PA: -Overall, 68.3% of parents perceived that adolescents’ school bags were too heavy to carry to school. This parental perception differed by adolescents’ mode of transport to school (active/motorized/combined: 35.1%/78.4%/68.8%, *p* < 0.001). -Adolescents perceived that their school bags were too heavy to carry to walk (57.8%) or cycle (65.8%) to school. -Adolescent perceptions differed by mode of transport to school (for walking (active/motorized/combined): 30.9%/69.2%/55.9% agree, *p* < 0.001; for cycling: 47.9%/72.8%/67.7%; *p* < 0.001).
Mandic, S. et al. [48], New Zealand, Health Research Council of New Zealand Emerging Researcher First Grant, National Heart Foundation of New Zealand, Lottery Health Research Grant, University of Otago Research Grant and Dunedin City Council.	The purpose of this study was two-fold: (a) to compare parental perceptions of walking versus cycling to school in an urban setting; and, (b) to examine if those parental perceptions of motivations for, and barriers to, walking and cycling to school differed based on distance between adolescents’ home and their school.	Cross-sectional study/socio-ecological model.	Parents (n = 341; age: 47.5 ± 5.2 years; 77.1% females; Maori = 6.6% and Pacific = 1.5%) completed a survey about their adolescent’s (age: 13–18 years; 48.1% boys) school travel and their own perceptions of walking/cycling to school.	Parents completed the BEATS Study Parental Survey. The geocoded home address was also used to calculate distance from home to adolescents’ school based on the shortest path on a connected street network using geographic information system (GIS) network analysis. Participants were categorised into three groups according to distance to school as “walkable” (2.25 km), “cyclable” (>2.25–4.0 km), and “beyond cyclable” (>4.0 km). The 4-point or 7-point Likert scale data were analysed as continuous variables with paired *t*-tests to compare parental perceptions of walking versus cycling to school. One-way ANOVA with Scheffe post hoc multiple comparisons or, when the assumption of homogeneity of variance was violated, Tamahane’s T2 test for comparisons across three distances to school categories were used. Comparison across two distances to school categories were conducted using an independent *t*-test or Mann-Whitney U test.	Common modes of transport to school differed significantly across the ‘walkable’/’cyclable’/’beyond cyclable’ categories (car passenger: 25.7%/40.5%/60.6%; public/school bus: 5.5%/15.4%/28.4%; walking: 66.2%/28.2%/1.2%; cycling: 0.0%/7.7%/0.5%; all *p* < 0.001). Barriers to PA: Compared to walking, parents perceived cycling to school to be less important (walking/cycling: 87.5%/62.5%), with less social support from parents (46.2%/17.1%), peers (20.6%/4.8%), and school (24.5%/12.4%), less interest from adolescents (48.5%/31.9%), fewer cycle paths (26.5%) versus footpaths (65.0%), and more safety concerns (35.0%/64.6%; all *p* < 0.001).
Pengpid, S. & Peltzer, K. [49], Cook Islands, Kiribati, Samoa, Solomon Islands, Tonga, Tuvalu and Vanuatu, WHO and Ministries of Education and Health in each country.	The aim of this investigation was to estimate the relationship between parental involvement and health behaviour and mental health among school-going adolescents in seven Pacific Island countries.	A secondary analysis of cross-sectional data using a two-stage cluster sampling study design/NR.	The sample included 10,968 school-going adolescents (mean age 14.1 years, SD = 1.4) from Cook Islands (overall response rate = 84%, n = 1274), Kiribati (85%, 1582), Samoa (79%, n = 2418), Solomon Islands (85%, n = 2211), Tonga (80%, n = 943), Tuvalu (90%, n = 943), and Vanuatu (response rate = 72%, n = 1119).	The “Global School-based Student Health Survey”(GSHS) comprises ten modules on various health behaviours, protective factors, and demographics. Pearson Chi-square statistics for categorical variables and ANOVA for continuous variables to calculate differences in proportion.	Overall, only 14.1% of the participants met daily PA recommendations, ranging from 10.8% in Vanuatu to 19.7% in Cook Islands. Facilitators for PA: -Parental involvement covered 4 areas: supervision (ranging from lowest 23.0% in Kiribati to highest 41.7% in Samoa), connectedness (15.3% in Kiribati to 32.6% in Samoa), bonding (18.8% in Tuvalu to 36.2% in Cook Islands), and respect of privacy (40.9% in Solomon Islands to 86.8% Tuvalu). -Higher parental involvement scores were positively correlated with meeting physical activity recommendations. Potential physical activity promoting strategies identified: -Parenting support programs, such as health-promoting strengthening activities for parents and children, should be supported in the study countries, to eventually improve health promotion targets.
Sheridan, S.A. et al. [55], Vanuatu, European Union’s Seventh Framework Program and the Australian Government’s NH&MRC-European Union Collaborative Research Grants.	This paper examined the perspectives of youth in Vanuatu on essential health needs in the context of the post-2015 development agenda, to make these concerns more visible for their communities, stakeholders, and health policy decision makers.	Qualitative study/NR.	The sample included twenty 17 year old secondary school students in Vanuatu.	Two focus group sessions, each consisting of 5 male and 5 female participants. A deductive thematic analysis was conducted.	Barriers to PA: -Local village leaders were often not reinforcing the government’s health promotion activities, and were seen to disregard the importance of PA.
Tuagalu, C. [50], Samoa, NR.	The research questions were (1) What are Samoan people’s perceptions and experiences of physical activity? (2) What barriers make it less likely that they will participate in physical activity? (3) What factors would make it easier for Samoan people to participate in physical activity?	Cross-sectional study/NR.	The sample included 801 participants from Samoa aged between 13–50 years, with 76% of the sample in the 13–18 years age group and two thirds (66%) were females.	The survey included questions that explored perceptions about physical activity, health, barriers to physical activity, sources of encouragement, and demographic trends. The data was analysed descriptively using Excel 2007 and SPSS.	Barriers to PA: -The participants reported that cultural (family, housework, and church), environmental (e.g., village curfew restrictions, safety—particularly from dogs, lack of footpaths), and discomfort (e.g., boring, too much effort) barriers were most likely to affect their participation in physical activity. Facilitators for PA: -The participants reported that their friends, school, church, doctor, partner, and village were a main source of encouragement to being physically active. -Most participants had a positive attitude towards physical activity, and more than half of them wanted to be more active.
Vancampfort, D. et al. [51], Kiribati, Samoa, Solomon Islands, Tonga, Vanuatu, Fiji and Tuvalu, no funding.	The study identified PA correlates including demographic variables (age, gender), policy related variables (e.g., provision of physical education classes), socio-environmental factors (e.g., food insecurity as a measure of proxy for socio-economic status, parental support, bullying), health behavior related variables (e.g., smoking, alcohol use, diet pattern), and health-related variables (obesity) among adolescents aged 12–15 years living in a LMIC and who participated in the Global school-based Student Health Survey (GSHS).	Cross-sectional study/NR.	The final sample consisted of 142,118 adolescents aged 12–15 years with a mean (SD) age of 13.8 (1.0) years, and 49.0% were girls. From the total sample, data from the following PICTs were included: Kiribati (collected in 2011; 85% participation rate), Samoa (2011; 79%), Solomon Islands (2011; 85%), Tonga (2010; 80%), Vanuatu (2011; 72%), Fiji (2016; 79%), and Tuvalu (2013; 90%) only.	Data from the Global school-based Student Health Survey (GSHS) were analysed. A multivariable logistic regression analysis was employed to assess the association between each correlate (exposure) and adequate PA (outcome). The analysis was adjusted for age, sex, and food in-security (proxy of low socioeconomic status). The association of age, sex, and food security with adequate PA was assessed with a model that mutually adjusted for these three variables.	Barriers to PA: -Adolescents with food insecurity (OR = 0.85; 95% CI = 0.80–0.90), low parental support/monitoring (OR = 0.68; 95% CI = 0.62–0.74), no friends (OR = 0.80; 95% CI = 0.72–0.88), and who experienced bullying (OR = 0.93; 95% CI = 0.86–0.99) were less likely to have adequate levels of PA. Facilitators for PA: Boys (OR = 1.64; 95% CI = 1.47–1.83) and those who participated in physical education for ≥5 days/week (OR = 1.12; 95% CI = 1.10–1.15) were more likely to meet PA guidelines.
Vargo, D. et al. [52], Samoa, US Department of Agriculture National Research Initiative	To describe a serious public health hazard in American Samoa that may plague other jurisdictions that tolerate a significant free-roaming dog population.	Cross-sectional questionnaire/NR	A survey of 437 adolescents (13–18 years; 220 males and 217 females) documented their experiences regarding unprovoked dog attacks.	The survey was designed to measure knowledge, attitudes, and practices regarding nutrition and exercise. Chi-square tests were performed using SigmaStat 3.1.	Barriers to PA: -About one-third of adolescents reported having been bitten by a dog between September 2010 and May 2011. About 10% of males and 17% of females attributed the fear of being bitten as a factor preventing them from being physically active. -Only “lack of time” and “lack of energy” elicited a greater number of responses than did the fear of being bitten.
Waqa, G. et al. [56], Fiji, Welcome Trust (in the UK) and the Fiji Health Sector Improvement Project of the Ministry of Health (MoH) of Fiji,	This paper describes the process evaluation for the Healthy Youth Healthy Communities project, undertaken in Fiji between 2006 and 2008. Process evaluation is important to determine whether the intervention was implemented as planned; to describe the intervention activities in terms of dose, frequency, and reach; and to identify any barriers to implementation.	Process evaluation/NR	A data collection proforma was developed to collate information about intervention planning and delivery activities (a description of the activity), processes (how the activity was conducted), dose (scale/duration of the activity), reach (how many and type of people involved in the activity), frequency (how often the activity was conducted), and associated resource use (for use in a subsequent economic evaluation). These data were supplemented by intervention reports, meeting minutes, correspondence, and communication between the research team staff and other personnel involved. A study manager, project coordinator and four research assistants (RAs) collected the data and completed the proformas.	Data were entered into an Excel database: more than 600 entries were recorded throughout the 2-year duration of intervention activities. Thematic analysis according to the four objectives of the Healthy Youth Healthy Communities project was conducted.	Facilitators for PA: -Walking, traditional dance, and aerobics. -Physical activity was often integrated successfully into the promotion of other strategies, such as the consumption of a healthy breakfast, fruit, vegetables, and water, especially during athletics season.

Abbreviations: body mass index (BMI), community based participatory research (CBPR), confidence interval (CI), not reported (NR), number (n), odds ratio (OR), overweight/obese (ow/ob), physical activity (PA), research assistants (RAs).

**Table 2 ijerph-19-11554-t002:** MMAT quality appraisal results.

Author (Year)	1. Qualitative					4. Quantitative					Quality of Studies (%)
	1.1 Approach Appropriate to the Research Question	1.2 Data Collection Methods Adequate	1.3 Findings Adequately Derived from Data	1.4 Interpretation of Results Substantiated by Data	1.5 Coherence between the Data Source, Collection, Analysis, Interpretation	4.1 Sampling Strategy Relevant to the Research Question	4.2 The Sample Representative of the Population	4.3 Measurements Appropriate	4.4 Risk of Nonresponse Bias Low	4.5 Statistical Analysis Appropriate to the Research Question	
Collier [57]						Yes	Yes	Yes	No	Yes	80
Curtis [53]	Yes	Yes	Yes	No	Yes						80
Fotu [44]						Yes	Yes	Yes	Yes	Yes	100
Hohepa [54]	Yes	Yes	Yes	Yes	Yes						100
Mandic [45]						Yes	Yes	Yes	Yes	Yes	100
Mandic [46]						Yes	Yes	Yes	Yes	Yes	100
Mandic [47]						Yes	Yes	Yes	Yes	Yes	100
Mandic [48]						Yes	Yes	Yes	Yes	Yes	100
Pengpid [49]						Yes	Yes	Yes	Yes	Yes	100
Sheridan [55]	Yes	Yes	Yes	Yes	Yes						100
Tuagalu [50]						Yes	Yes	No	Yes	No	60
Vancampfort [51]						Yes	Yes	Yes	Yes	Yes	100
Vargo [52]						Yes	Yes	Yes	Yes	Yes	100
Waqa [56]						Yes	Yes	No	Yes	No	60

**Table 3 ijerph-19-11554-t003:** Barriers (B) and facilitators (F) to physical activity and sport in a socioecological framework and each study.

Socio-Ecological Level	Description of Barrier/Facilitator	Study Reference, Income (High (H), Middle to High (MH), Low to Middle (LM) and Low (L)) and Reporting Barrier (B)/Facilitator (F)														Total Number of B and F	
		Collier [57], H	Curtis [53], H	Fotu [44], MH	Hohepa [54], H	Mandic [45], H	Mandic [46], H	Mandic [47], H	Mandic [48], H	Pengpid [49], MH, LM, L	Sheridan [55], LM	Tuagalu [50], LM	Vancampfort [51], MH, LM, L	Vargo [52], LM	Waqa [56], MH	B	F
Individual																	
	Time		B											B		2	
	Play		F														1
	Fun		F		F												2
	Preferential activity				F												1
	Positive self-esteem		F														1
	Feelings of achievement				F												1
	Physical health benefits				F												1
	Physical appearance				F												1
	Mood, confidence				F												1
	Low interest						B		B							2	
	Discomfort/sweaty					B						B				2	
	Energy													B		1	
Interpersonal																	
	Friends/PA partner	F			F				B			F				1	3
	Money		B													1	
	Transportation		B													1	
	Bullying/no friends												B			1	
	Walking to school					F											1
	Parental support and involvement								F	F		F	F				4
	Connection to cultural practices (e.g., church, traditional dance, walking)											BF			F	1	2
	Heavy school bags							B								1	
Community																	
	Medical/doctor support											F					1
	Address socio-cultural factors			F													1
	Community leaders’ support		F								F	F					3
	Weather conditions	B				B										2	
	Minimal indoor facilities for PA/sport	B														1	
	Distance for active transport				B		F									1	1
	Safety		B		B	B			B			B				5	
	Fear of dogs and dog bites											B		B		2	
	Inadequate footpaths/cycle paths								B			B				2	
	Structure of PA/sport opportunities	B			B											2	
Policy/institutional																	
	Availability and continual access to PA/sport programs				F												1
	PE in schools 5 days/week												F				1
	Food insecurity												B			1	
	Village curfews											B				1	

## Data Availability

Not applicable.

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
