# Peer review of "The Barriers to and Facilitators of Physical Activity and Sport for Oceania with Non-European, Non-Asian (ONENA) Ancestry Children and Adolescents: A Mixed Studies Systematic Review"

_ijerph, 2022, doi:10.3390/ijerph191811554_

Round 1

Reviewer 1 Report

This review aimed to synthesize existing quantitative and qualitative literature regarding 21 barriers to and facilitators for physical activity (PA) and sport in Oceania with Non-European, Non-Asian (ONENA) youth. Literature was systematically searched to include studies reporting barriers to or facilitators for PA and/or sports participation among ONENA children and adolescents aged 0–18 years residing in Pacific Island Countries and Territories (PICT) and Australia. The authors pointed out that programs that offer PA and sport participation options with embedded self-determination theory (SDT)-informed strategies for all family members, while being accessible through existing transport and related social/ cultural/physical infrastructure, and committed to communities through formal partnerships should be needed to enhance PA and sport participation of ONENA youth residing in 10 PICT. Study motivation and background that were described in this manuscript aligned with study goal.

The authors discussed their findings based on barriers to and facilitators of PA categorized into four levels: Individual, Interpersonal, Community and Policy level. Barriers and facilitators cannot be easily determined by certain levels or factors since they are interrelated to each other across different levels. For example, social determinants of health (SDoH), which contribute to wide health disparities and inequities, could be considered to enhance PA and sport participation. It would be helpful if the authors can provide further discussions related to this point to make proper directions and guidance in this area.

Author Response

Dear Reviewer 1,

Please see response to your feedback attached.

Thank you,

Author

Author Response

Dear Reviewer 2,

Please see attached document for responses to your feedback.

Thank you,

Authors
